# Avian Diversity Responds Unimodally to Natural Landcover: Implications for Conservation Management

**DOI:** 10.3390/ani13162647

**Published:** 2023-08-16

**Authors:** Rafael X. De Camargo

**Affiliations:** 1Laboratoire Chrono-Environnement, UMR-CNRS 6249, Université Franche-Comté—UFC, 25030 Besançon, France; rafael.x.camargo@gmail.com; 2TRANSBIO Graduate School, Université Bourgogne Franche Comté—COMUE UBFC, 25000 Besançon, France

**Keywords:** birds, species richness, biodiversity, natural landcover amount, habitat amount, range maps, breeding bird survey, species pool, peaked species–habitat relationship

## Abstract

**Simple Summary:**

Human activities have imposed unprecedented changes on more than half of the Earth’s terrestrial surfaces. Biodiversity losses will ensue. Considering patchy landscapes (e.g., different landcover types, from completely “unnatural” to “natural” systems), what should be the shape of the relationship between the total number of species as a function of the proportion of natural landcover? Current theoretical and empirical studies have insisted on monotonic increasing relationships, implying species linearly declining as natural covers are converted to human-dominated areas, ignoring non-linear unimodal ones. We addressed this issue, offering potential explanation of which factors may be linked to hump-shaped relationships between avian diversity and gradients of natural landcover in landscapes of different sizes (25–900 km^2^) in Ontario and New York State. We showed that the hump-shaped pattern is consistent across spatial scales and bioclimatic regions, where diversity reaches its maximum of around 40–60% of natural landcover. Pragmatic conservation actions aiming to mitigate biodiversity loss from land-use modifications should focus on alleviating environmental stress in intensively used areas while managing efficiently the ones holding moderate proportions of natural habitats.

**Abstract:**

Predicting species’ ecological responses to landcovers within landscapes could guide conservation practices. Current modelling efforts derived from classic species–area relationships almost always predict richness monotonically increasing as the proportion of landcovers increases. Yet evidence to explain hump-shaped richness–landcover patterns is lacking. We tested predictions related to hypothesised drivers of peaked relationships between richness and proportion of natural landcover. We estimated richness from breeding bird atlases at different spatial scales (25 to 900 km^2^) in New York State and Southern Ontario. We modelled richness to gradients of natural landcover, temperature, and landcover heterogeneity. We controlled models for sampling effort and regional size of the species pool. Species richness peaks as a function of the proportion of natural landcover consistently across spatial scales and geographic regions sharing similar biogeographic characteristics. Temperature plays a role, but peaked relationships are not entirely due to climate–landcover collinearities. Heterogeneity weakly explains richness variance in the models. Increased amounts of natural landcover promote species richness to a limit in landscapes with relatively little (<30%) natural cover. Higher amounts of natural cover and a certain amount of human-modified landcovers can provide habitats for species that prefer open habitats. Much of the variation in richness among landscapes must be related to variables other than natural versus human-dominated landcovers.

## 1. Introduction

Conversion of natural landcover into human-modified landscapes may lead to rapid and potentially irreversible biodiversity changes at local scales worldwide [1,2,3]. Currently, more than half of the Earth’s surface has some degree of modification, leaving most areas with mixtures of natural habitats, agricultural lands, and areas more intensively utilised by humans [4,5]. Intensification of land use often leads to the extirpation of natural habitats, causing local extinctions and potentially overall species losses. International initiatives have recently called for half terrestrial-realm protection to halt species extinctions and biodiversity erosion by 2050 [6,7]. This target seems to be reasonable and necessary. Yet it has not been based on empirical evidence, since model predictions of how biodiversity responds to human modification of landscapes remain largely inaccurate [8,9,10].

A general pattern for the species richness–natural landcover relationship could guide conservation practices on how much habitat should be protected, conserved, and restored [11,12,13]. Most approaches to modelling spatial variation in species richness are built on classic species–area relationships (SAR models). Classic SAR models (S = cAz, where S = species diversity, A = area, and c and z are empirical constants) assume that species richness increases as a power function of area [14], and they have been widely applied in global assessments of biodiversity to estimate species loss from habitat loss [15,16,17]. Yet SARs have been shown consistently to overestimate species losses in practice [18]. This happens because these models ignore the fact that natural landcover that is transformed by human activity is not necessarily lost. It may provide habitats for different species, defying the overall SAR modelling premises that more natural habitat leads to more species diversity in landscapes.

More recent models have attempted to make better predictions by showing that species distinguish among multiple landcover types in human-modified landscapes. Called multi-habitat SAR or countryside SAR models, these approaches assume that landscapes contain different habitat types resulting from habitat conversion and that identifiable guilds of species respond differently to the amount of each habitat type [19]. They usually perform better than classic SAR models [20] since they distinguish only between the amounts of natural and human-modified covers in a landscape [18,21]. Despite that, multi-habitat equations are essentially twisted power functions of the area and mostly predict richness increase as a function of habitat in a monotonic fashion [22,23]. Moreover, for a general model to be useful for applied purposes, it would have to capture a high proportion of the spatial variability in richness (e.g., high R^2^), and it must not vary among geographic regions (see, for example [22,24]).

Non-monotonic relationships of richness versus natural landcover have been less frequently documented in the literature but are not rare. Habitat variability may explain these patterns at the landscape level [25]. Hump-shaped richness as a function of gradients of natural landcover amounts (e.g., 0–100% human-dominated to natural habitats) has been documented in plants [26], birds [27,28,29], and invertebrates [29]. Desrochers et al. [27] observed that avian species richness peaks as a function of the natural landcover in Southern Ontario, Canada. Building on their work, De Camargo and Currie [28] empirically demonstrated that the peaked richness–natural landcover relationship is a result of (1) a relationship describing forest species richness increasing as a power function of forest amount, summed with (2) a peaked relationship of the richness of open-habitat species as a function of the amount of human-dominated landcovers. The authors conceptualised the Lost-Habitat SAR to explain peaked patterns, in which parts of human-dominated landcovers in highly anthropogenic landscapes are unavailable for some open-habitat species. Thus, a mixture of both forested- and human-dominated habitats accommodates a higher diversity total number of species (see also [23,30,31,32]). As a result, species richness may increase with habitat heterogeneity at moderate levels of pressure [33,34], declining when human activities cumulates in the environment. In turn, richness may not increase monotonically with the amount of natural habitat in landscapes [29,29,35,36].

Here, we test the overarching hypothesis that the proportion of natural landcover amounts drive species richness at the landscape level and across large regions. Specifically, we test a series of predictions on the underlying factors driving the peak relationships between avian species richness and natural landcover that were observed in our previous work in Southern Ontario, Canada (Table 1). We found hump-shaped rather than monotonic positive curves consistently describing the richness–natural landcover relationships within multi-scale (25–900 km^2^) landscapes across New York State, NY, USA, and Ontario, ON, Canada. We discuss underlying mechanisms of hump-shaped richness–natural cover relationships and their conservation implications.

**Table 1 animals-13-02647-t001:** Predictions of the hypothesis that avian species richness peaks at intermediate levels of natural landcover amounts.

Predictions	Rationale	Test	Expected Results
P1. Richness–natural landcover relationship is peaked rather than monotonically positive, independently of spatial grain size (e.g., landscape size) and geographic region	SAR power functions preclude hump-shaped richness–natural landcover relationships [28]. Would the peaked relationship be a common pattern elsewhere and across spatial scales?	Generalised linear models/autoregressive models Landcover model richness = f(proportion of natural landcover) *	- Richness peaks rather increase monotonically at intermediate levels of natural landcover within multiple-size landscapes and geographic regions with similar biomes
P2. Collinearity between natural landcover and temperature may drive observed peaked relationships between richness and natural landcover	Agricultural suitability and temperature in Ontario both show strong south–north gradients, but with (potentially) opposite effects on birds (Figure 1). It is possible that avian richness is low in Southern Ontario due to low forest cover and low in Northern Ontario (where forest cover is high) because of low temperatures	Generalised linear models/autoregressive models * Landcover, temperature model richness = f(proportion of natural landcover, temperature) *	- Richness–natural landcover is peaked even after accounting for temperature
P3. Habitat heterogeneity (i.e., variety of landcover types) within landscapes may explain the peaked relationship between richness and natural landcover	It has been hypothesised that a greater number of landcover types may increase species richness within a given landscape. Desrochers et al. [27] showed that different landcover types partially explain the peaked pattern observed in landscapes of Ontario	Generalised linear models/autoregressive models * Full model richness = f(proportion of natural landcover, landcover variety, temperature) *	- Landcover variety may override effects of natural landcover amounts on species richness in full models

* Models controlled for effort and regional richness (pool of species) and fitted with varying landscape sizes (25–900 km^2^) in New York State and Ontario (i.e., study areas share similar biomes).

**Figure 1 animals-13-02647-f001:**
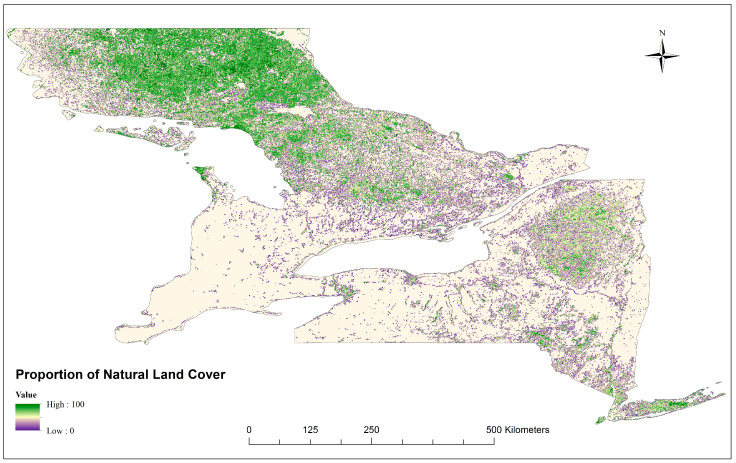
Gradients of proportion natural landcover (pNLC) covering the study area according to the global 1 km consensus landcover dataset [37]. The projection is the WGS84 datum.

## 2. Methodology

### 2.1. Study Region and Species Richness

The study region includes Southern Ontario, ON, Canada (~200,000 km^2^), and New York State, NY, USA (~125,400 km^2^) (Figure 1). Both areas contain mixed hardwood forests in lowland areas and boreal forests in areas on Precambrian granite shield and at higher elevations. Forest clearing in the area occurred principally during the 17th- to 19th-century European settlement of the area [28].

To calculate landscape-level avian species richness in this study, we used species distribution data from the Ontario Breeding Bird Atlas (OBBA [38]) and the New York State Breeding Bird Atlas (NYBBA [39]). Both atlases were based on systematic surveys conducted between 2000 and 2005. Both used experienced birders to identify the breeding bird species occurring within each quadrat. Sampling was carried out over several years and was designed to sample all habitats in a grid cell (hereafter, grid cells, quadrats, and landscapes are interchangeable terms and represent the sampling units). Since the goal was to find all species breeding in each quadrat, we treated species not observed in a quadrat as being truly absent [40]. Richness in a quadrat represents the total number of species observed breeding in that quadrat during one or more years in the five-year period. For both sets of atlas data, we excluded wedge-shaped UTM quadrats and quadrats with more than 10% lake area to minimise “area” variation among quadrats.

Species richness estimates may depend upon the size of the sampling unit [41]. Hence, we tallied bird species richness in three landscape sizes: 25, 100, and 900 km^2^. We considered these spatial grain sizes relevant for conservation and management purposes since conservation decisions are often made at grains of the order of 10–1000 km^2^, reflecting planning, zoning, and other landscape or conservation management ordnances [42]. The NYBBA sampled birds on a 5 × 5 km grid; the OBBA used a quadrat size of 10 × 10 km. To compare richness between atlases, we resampled the NYBBA data at the 10 × 10 km quadrat size (same cell size as the OBBA). We calculated richness by counting the number of unique species’ presences from the original survey quadrats within each new 100 km^2^ grid cell in New York State. We also compared survey richness in 30 × 30 km grid cells in both regions, following the same resampling procedures. The total numbers of grid cells for each landscape size analysed were 4822 for NY at 5 × 5 km; 985 and 1075 for ON and NY, respectively, at 10 × 10 km; and 251 covering ON and NY at 30 × 30 km.

### 2.2. Independent Variables

#### Main Predictors: The Amount of Natural Landcover, Temperature, and Landcover Variety

SAR models predict overall richness increases as the proportion of natural areas increases [43,44]. To estimate the proportion of natural landcover (pNLC) within landscapes, we used a global consensus landcover dataset [37] (Figure 1). The dataset was composed of 12 landcover classes, observed at a spatial resolution of 30 arc-seconds (~1 km^2^), and derived from satellite imagery from 1999 to 2006. The landcover classes were: 1. evergreen/deciduous needleleaf trees, 2. evergreen broadleaf trees, 3. deciduous broadleaf trees, 4. mixed wood/other trees, 5. shrubs, 6. herbaceous vegetation, 7. cultivated and managed vegetation, 8. regularly flooded vegetation, 9. urban/built-up, 10. snow/ice, 11. barren, and 12. open water. The proportion of pixels in each landcover class was determined within each quadrat (0–100%). To obtain the pNLC in each of our sampling units, we summed classes 1–6 and 8.

Temperature strongly correlates positively with species richness patterns at broad scales in mid to high latitudes [45,46]. Hence, we used Mean Annual Temperature (MAT) as a predictor of avian richness for each grid cell at different spatial scales. Temperature data were obtained from the WorldClim database, aggregated across a target temporal range of 1970–2000 at 1-km^2^ resolution Figure 2 [47]. Although we could have used other temperature variables (e.g., breeding season temperature), different temperature metrics are very strongly collinear at broad spatial scales, and one would expect similar relationships between richness and most measures of temperature.

It has been hypothesised that species richness may increase with the number of different habitat types at the landscape level (i.e., habitat heterogeneity) [34,44], and that may lead to humped-shape richness-habitat relationships [27]. To test this hypothesis, we included the number of different landcover classes [37] as a measure of landcover variety (LCV) in our models. We also fitted the models using the diversity of landcover types, based on the Shannon index. Since the results were qualitatively similar using the number of landcover classes and the diversity of classes (see also Desrochers et al. [27]), we present model results fitted with the former.

### 2.3. Statistical Analysis

We used Ordinary Least Squares (OLS) regression models to relate species richness to the predictors and covariates described above. For the full model, we compared the partial regression coefficients and statistical significance within a model including all three predictors (pNLC, MAT, and LCV) and the covariates listed below. We fitted survey richness as quadratic functions of these variables to allow for non-linearity. We considered a relationship between richness and landcover to be peaked if the (a) the quadratic term in the regression was significant and (b) the hypothesis that the maximum or minimum of the curve fell outside the observed range of pNLC could be significantly rejected, as determined by an MOS test [48]. We presented statistical results for the full model and for the average of all supported models in a multi-model inference framework [49]. For the multi-model inference analysis, we compared the averaged partial regression coefficients from all models within a 95% confidence set. Similar results from both methods give us extra confidence that our estimates of relative importance are meaningful [50].

We dealt with the collinearity between temperature and pNLC in different ways. First, we visually analysed the spatial distribution of species richness across both study areas and how temperature influenced pNLC through simple linear regressions. Then, we analysed whether temperature and pNLC were correlated beyond the arbitrary threshold (Pearson’s correlation >0.7) [49]. Finally, we analysed the richness–natural landcover relationships where collinearity between temperature and pNLC is lowest by sub-setting the data in two ways: (a) small temperature range and maximum variability in pNLC (5 °C ≤ MAT ≤ 10 °C, see Appendix A) and (b) in the coolest and warmest places in both study areas (e.g., Figure 2—Boreal Shield, ON, and Adirondacks, NY State).

#### Model Covariates: Sampling Effort and the Size of the Regional Species Pool

The influence of sampling effort on estimates of total species richness has been known for a long time [51,52]. For the NYBBA, atlassers were assigned to survey one or more NYBBA quadrats and were expected to spend at least 8 h in each 25 km^2^ block, visiting each habitat present and recording at least 76 species. We excluded quadrats with efforts ≤8 h, totalising 4822 quadrats of 5 × 5 km with a median effort ≅10.5 h. For resampling, we summed up original sampling efforts from 5 × 5 km that were encompassed into 100 and 900 km^2^ landscapes.

For the OBBA surveys, each volunteer was assigned to search a specific 100 km^2^ quadrat as completely as possible for evidence of all species breeding therein. Volunteers were instructed to search for regionally rare species. We excluded 7 quadrats in which sampling effort was much higher than in all other quadrats (1200 h, vs. 10–430 h in other quadrats) because these two quadrats had very high leverage in the regression models. That led to a total of 985 quadrats of 10 × 10 km with a median effort ≅45 h. For the 30 × 30 km resampling, we summed up original sampling efforts from 10 × 10 grids that were encompassed into 900 km^2^ landscapes. Since the original OBBA quadrats were four times larger than the NYBBA quadrats, the effort per unit area was relatively similar in the two atlases, making the data comparable in terms of sampling effort.

The richness of local communities is limited primarily by local factors, but it may also be influenced by the size (richness) of the regional pool of species [53,54]. The richness of the regional pool can also be estimated by superimposing species’ range maps resolved at fairly coarse spatial grains (e.g., ~10^4^ km^2^) [55] and tallying the number of ranges that overlap each landscape. Range maps were obtained from the BirdLife International World Bird Database [56]. We overlaid species’ ranges on the 25 km^2^ (NYBBA) and 100 km^2^ (OBBA) quadrats. We resampled New York at 100 km^2^ and both Ontario and New York with 30 × 30 km grid cells (900 km^2^). Hence, we calculated the richness of the pool of birds occurring at the regional level for each landscape containing richness from atlases.

Spatial autocorrelation can affect model coefficients in spatial analyses [57]. In our data, avian richness is spatially autocorrelated (Moran’s I = 0.10 in 5 × 5 km, 0.12 in 10 × 10 km, and 0.14 in 30 × 30 km landscapes at the nearest distance class and declines with distance in each respective dataset). This may be related to spatially structured landscape features, such as natural landcover, habitat heterogeneity, and/or climate variables (e.g., temperature). Thus, we fitted simultaneous autoregressive error (SARerr) models, as proposed by [58], in R (“spatialreg” package, “errorsarlm” function). To show how much variance explained in richness is due to spatial autocorrelation, we present autoregressive models in the main text and equivalent results from OLS models in the Appendix A. Spatial data, including satellite images and climate raster files, were treated in ArcGIS, and all statistics were carried out in R, more specifically, using the following packages: stats, vegan, ggplot, reshape2, RColoBrewer, grid, lm.beta, polynom, fms, and AICcmodavg, to name a few of them [59].

## 3. Results

Supporting Prediction P1 (Table 1), bird species richness peaks rather than monotonically increasing with the increasing proportion of natural landcover within landscapes (Figure 3). Richness peaks at ~50–60% pNLC in Ontario and New York State. The shape of the relationship between avian richness and amount of natural landcover is similar across the 25 km^2^, 100 km^2^, and 900 km^2^ cell sizes; the strength of the relationships (R2) increases slightly with grain size (Figure 3). The hump-shaped relationship found in New York State is very similar to the richness-pNLC relation previously observed in 100 km^2^ landscapes of Ontario (see Figure 2a in Desrochers et al., 2011 [27]). MOS tests confirmed that the peaks of the polynomials fall within the range of the data (Appendix A).

Could the peaked relationship between avian richness and pNLC be due to collinearity between landcover and temperature (P2, Table 1)? Spatial autoregressive models showed that bird species richness in both regions is still a peaked function of amount of natural landcover after accounting for temperature in both regions (standardised coefficients in Table 2). Incorporating spatial autocorrelation increased the variance explained of richness models by 1–19%, where explained variance increases as landscapes become larger (Table 2 vs. Appendix A). Averaging of standardised coefficients obtained from multi-model inference analysis (Appendix A) showed similar result patterns to full models’ outcomes (Table 2); temperature and amount of natural landcover are the main drivers of species richness patterns across spatial grain sizes and in both geographic regions.

Avian richness derived from atlas data and from range maps varies spatially in different ways (Appendix A). The geographic variation of atlas richness is more spatially structured over short distances (maps in Appendix A), reflecting the finer spatial grain of the underlying species distribution data. The spatial structure of atlas richness resembles the variation in pNLC in the study area, especially in areas with reduced amounts of natural landcover, such as Southwestern Ontario and highly urbanised areas in New York State (cf. Figure 1 vs. Appendix A). In contrast, the spatial structure of range-map richness follows the temperature/latitudinal gradient (cf. Figure 2 and Appendix A). Total species richness from atlas data does not seem strongly dependent on the regional size (richness) of the species pool (Appendix A). Moreover, avian richness of the regional pool of species increases with temperature in both New York and Ontario (Appendix A). In contrast, atlas richness is a peaked function of temperature in both regions (Appendix A).

The partial regression coefficients are similar in NY and ON, despite the much lower collinearity between temperature and pNLC (data subset: 5 °C < MAT > 10 °C) (Table 2 main text vs. Appendix A). Moreover, the shape of the relationship remains peaked in the warmest and the coldest places of both study areas (Appendix A). Within these subsets of reduced temperature variation, pNLC still varies considerably (see Appendix A). These results are inconsistent with the proposition that the peaked relationship is solely due to collinearity between temperature and pNLC.

Bird species richness within landscapes had a weakly positive relationship with landcover variety (i.e., habitat heterogeneity) (Table 2, Figure 4). These results do not support the assumption that a variety of habitat types within landscapes (habitat heterogeneity) leads to more species diversity (P3, Table 1). The term increased the explained variance in avian richness by 1–3% (results not shown). Collinearity between landcover variety and the other independent variables used in the models was relatively weak (Appendix A), and the peaked relationship between richness and pNLC persisted after controlling for habitat heterogeneity (Table 2).

## 4. Discussion

Previous studies have shown that bird species richness peaks rather than increases as the proportion of natural landcover (pNLC, i.e., mainly forested areas) varies within landscapes of Southern Ontario, Canada [27,28]. In this study, we empirically tested predictions related to underlying factors that could help to explain the peaked pattern. We found that the peaked relationship (1) is also observed in similar ecosystems in another geographic region (New York State); (2) is somehow affected by temperature gradients, but not solely due to collinearity with climate; (3) is not explained by landcover variety; (4) is not due to variation in the size of the regional pool of species; and (5) is not dependent on spatial grains between 25 and 900 km^2^. Controlling for these potentially confounding variables does not lead to a monotonic positive relationship between richness and the amount of natural landcover. Perhaps surprisingly, species richness is not strongly related to the variation in natural landcover among landscapes. The peaked pattern has been largely ignored in current modelling approaches trying to predict biodiversity change from landcover changes. We address each of these points in turn.

Calculations of the effects of landcover conversion on biodiversity loss have led to notorious overstatements, e.g., “[…] current rates of extinction are about 1000 times the likely background rate of extinction” [61]. These extinction rates entirely based on SAR models [62] have been repeated to exhaustion even though they have been proven wrong [18]. If human-dominated landcovers were completely unavailable to species (i.e., from the typical binary power function species–area relationship), then there should be a monotonic positive relationship between species richness and the total area of remnants of natural areas, irrespective of spatial grain (see, for example, [63]). Hence, SARs have frequently been used to forecast species losses (e.g., the number of species extinct or threatened) from the removal of natural (usually forested) cover [61,64,65,66,67]. At coarse spatial grains and for large extents, those forecasts have greatly exceeded observed species losses [18,22,67,68]. The discrepancy is sometimes attributed to “extinction debt”: extinctions that are predicted to occur but have not had time to do so. The difficulty is that the concept of “extinction debt” assumes that the causal link between species extinction and habitat loss exists, despite data to the contrary. Further, without a specified time by which the debt will have been resolved, the idea is untestable. Nonetheless, species–area relationships are still commonly applied in conservation studies to predict loss of species as a function of habitat modification, assuming that extinction debts will be paid [61,65,67,68,69]. In contrast, if natural landcovers and human-dominated landcovers provide habitats for different sets of species [28,31], then conversion of some natural landcover to human-dominated covers may increase the richness of species that prefer human-dominated (often open, early successional) habitats to a greater extent than it decreases the richness of forest species. There may be no need to postulate extinction debt.

Most current multi-habitat models do not account for peaked relationships between species’ ecological responses (e.g., occurrences, richness) and species’ habitats. Countryside models have indeed outperformed most single-habitat models (SARs and derivations) and other multi-habitat models [20,23,31,70]. This is because they assume that species respond to the increase in species’ amount of habitat found within landscapes. However, their concept might be fundamentally flawed for three reasons: (1) not all landcover may be available to species in human-modified landscapes. For example, intensively used areas may not offer habitats for any species [28]; (2) richness does not vary strongly with the number of different classes of satellite-sensed landcover (Figure 4). Rather, the richness of different species sharing similar habitat requirements (e.g., amounts of open- and human-dominated landcovers) may respond equivalently to different landcover types (e.g., pasture, urban, abandoned fields, etc.) [71]; and (3) the conversion of natural habitat to human-dominated cover is not the only way that human activity may influence the ability of species to persist in a landscape (i.e., hunting, pollution, etc.).

Habitat loss per se may not be the biggest threat to biodiversity, except in landscapes with very little natural habitat [13,72]. Our relationships between species richness and the amount of natural landcover are not especially strong. However, richness increases monotonically and positively in landscapes with 50% or lower amounts of natural landcover (see Figure 3 in this study, Fahrig et al., 2013 [21]). Other factors must therefore play a major role in explaining diversity decline or act in combination with habitat loss to imperil species [73]. For instance, hunting practices have been linked to prehistoric [74] and modern [75] species extinctions. Yet hunting is poorly represented in assessments of threats to biodiversity [76]. Species losses have been related to pesticide use in agricultural landscapes [77,78,79,80]. Also, land-use intensity has emerged as a potential major driver of species declines worldwide [1,81,82,83,84]. Southeastern Ontario and Western NY State are heavily agricultural. Long Island has a high human population density. It is in these areas that survey richness is the lowest. The relationship between landscape-scale avian species richness and the amount of natural landcover is neither monotonically positive nor particularly strong. This suggests that the contention that “In general terms, the loss of biodiversity is caused by habitat loss…” [85] is an over-simplification, true mainly at the limit (i.e., again for landscapes holding <50% natural cover).

Climate (mainly temperature) drives species richness at coarser scales [86,87,88]. While the mechanisms underlying such patterns are contentious [89,90,91,92], range-map-derived richness of most species groups increases monotonically with mean annual temperature (MAT) and/or a moisture–heat interaction [86]. At a finer grain (e.g., 5–100 km^2^), the influence of temperature on species richness is less clear [86]. 

Avian range-map richness is less strongly related to the amount of natural landcover, and the relationship is monotonically negative (Appendix A). This is true in two independent areas (NY and ON) and in landscapes that vary in size from 25 to 900 km^2^. It seems unlikely that greater forest cover causes low range-map richness. Ranges circumscribe occupied and unoccupied areas, and landcover varies dramatically within individual species’ ranges (Figure 1). Range maps rarely exclude areas of absence within a species’ range (see any species’ range map on Birdlife International 2020). There is little reason to expect ranges to respond to landcover changes, except perhaps by contracting at range margins. A more likely explanation for the negative relationship is that there has been greater forest loss in warmer areas of New York and Ontario [46], where more species’ ranges overlap (Figure 1). We regard range-map richness as a measure of the number of species that could potentially occupy a landscape rather than a measure of landscape richness.

It has long been hypothesised that habitat heterogeneity may increase species diversity [93]. At the landscape level, empirical evidence remains scarce and debatable (see reviews in [25,94]). At coarser scales, studies have found that areas presenting high ranges of elevation, precipitation, vegetation, and other environmental features may promote the diversification of habitat or energy sources, which could in turn increase species richness [95,96,97]. In areas with very little natural habitat (e.g., intensive agricultural fields), our results are consistent with earlier work indicating that richness increases rapidly with the amount of natural habitat [28,33,98] because the landscape provides habitat for forest birds. Similarly, in areas with very little open habitat, the richness of open-habitat species increases rapidly when some forest is replaced with human-modified landcovers. This may not be a simple matter of landcover classes as surrogates for habitat heterogeneity. Many bird species share similar habitat requirements (e.g., >2–3 landcover classes), which could explain the fact that the number of different landcover classes was only a weak predictor of richness in both ON and NY (Figure 4). 

Finally, species richness is only one metric of the conservation value of landscapes. Conservation may focus on individual species, ecosystem services, carbon storage, spiritual values, or other properties [99]. We chose to focus here on species richness because it is a high-level integrated index that has a long conceptual and empirical history. Also, it should be noted that new modelling approaches using other biodiversity metrics (e.g., phylogenetic and genetic diversity metrics) have the potential to overcome the basic modelling shortcoming discussed in this study [100]. In parallel to this study, De Camargo et al. [71] examined the probabilities of occurrence of individual species in landscapes as functions of landcover. Their conclusions are broadly like those presented above.

## 5. Conclusions

Nature and humans form coupled systems presenting non-linear relationships [98]. Moderate pressures in terrestrial ecosystems may boost species diversity, while intensively used land will cause species losses. Currently, modelling approaches have ignored these unimodal relationship patterns, failing to guide efficient conservation management at the landscape scale across large regions. Further studies should focus on identifying human pressures that may cause species to decline at lower values of the proportion of natural landcover at local scales across large spatial extents. A general macroecological framework on species’ ecological responses to cumulative pressures in the environment may better predict diversity changes due to habitat loss than current species–area relationship models.

## Figures and Tables

**Figure 2 animals-13-02647-f002:**
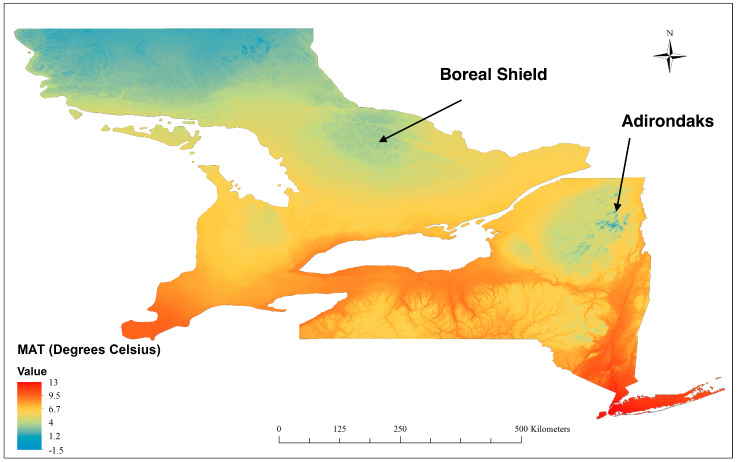
Mean Annual Temperature (MAT) covering the study area according to WorldClim [47]. The projection is the WGS84 datum. Higher-elevation areas in Southern Ontario (Canadian Shield) and New York (Adirondack Mountains) are both cooler and on Precambrian granite.

**Figure 3 animals-13-02647-f003:**
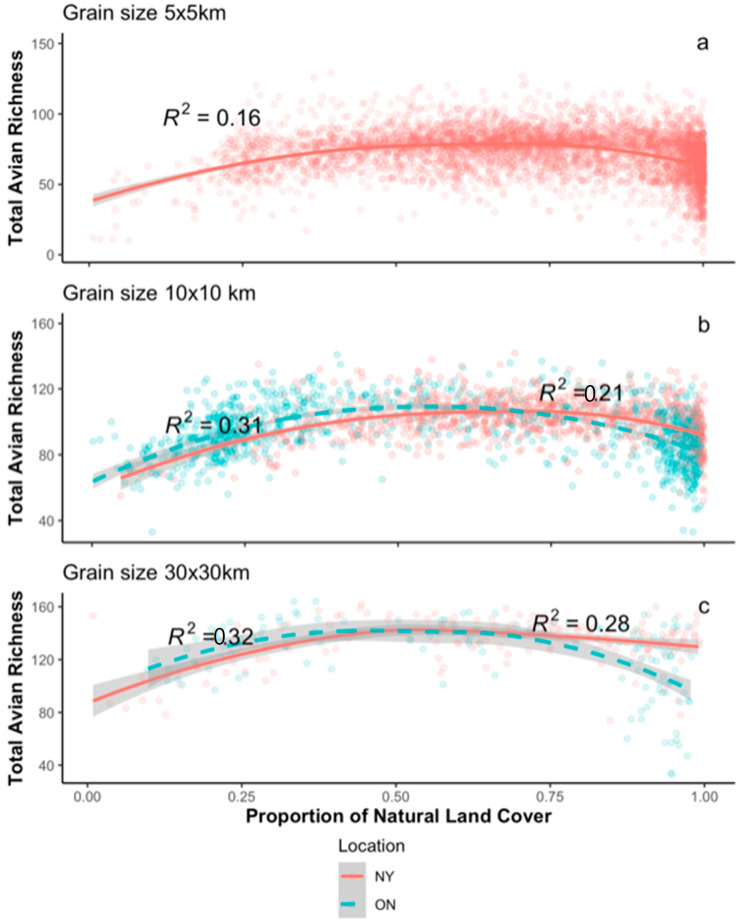
Relationships between bird species richness and landcover in grid cells covering Southern Ontario and New York State at different spatial grain sizes (5 × 5 km, 10 × 10 km, and 30 × 30 km). R^2^ represents the goodness of fit of second-degree polynomial OLS regression models. Survey richness peaks at 62% natural cover in 5 × 5 km quadrats in NY (n = 4822), 64% in 10 × 10 km quadrats in NY (n = 1075), 64% in 30 × 30 km quadrats in NY (n = 165), 54% in 10 × 10 km quadrats in ON (985), and 50% in 30 × 30 km quadrats in ON (n = 138).

**Figure 4 animals-13-02647-f004:**
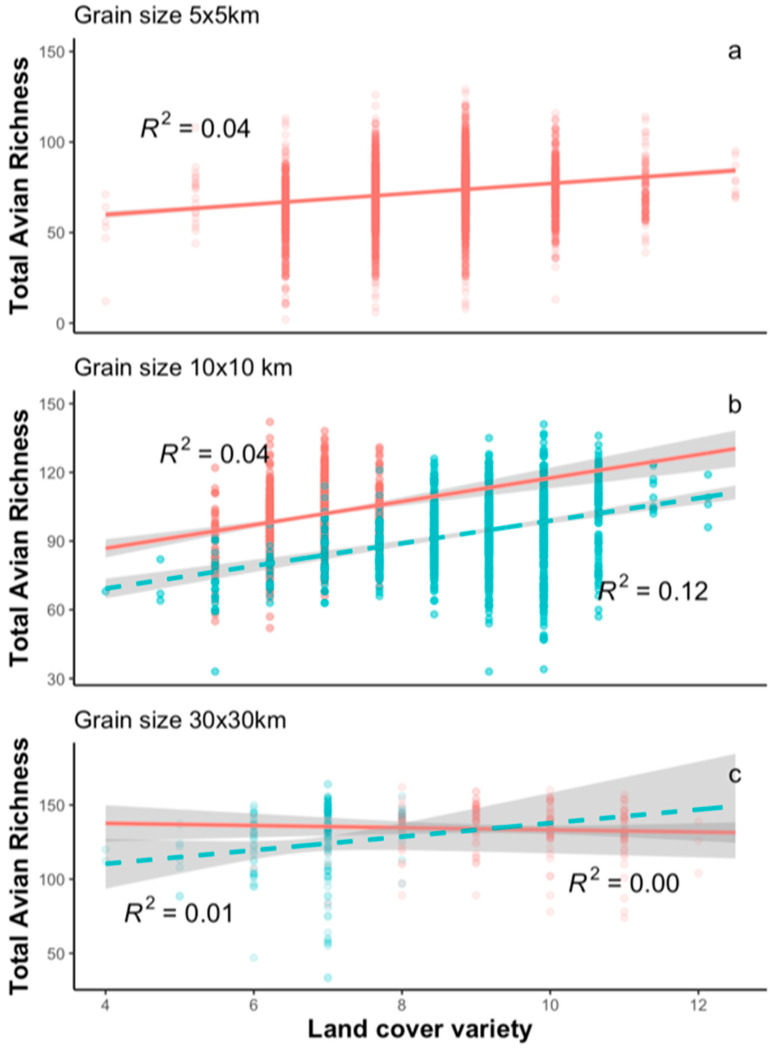
Relationships between bird species richness (derived from atlas data) and landcover variety in landscapes of different spatial grain sizes in both study areas (ON and NY State).

**Table 2 animals-13-02647-t002:** Autoregressive model results for predictors of total avian richness in landscapes of three different sizes in Southern Ontario (ON) and New York State (NYS). Predictors included in full models: annual mean temperature (MAT), the proportion of natural landcover (pNLC), and landcover variety (LCV), and covariates: sampling effort (E) and the size (richness) of the regional pool of species (Pool). *n* represents the number of landscapes in each study area. AICc is the Akaike information criterion corrected for sample sizes [60]. *Nagelkerke pseudo-R*^2^ is the equivalent to regular adjusted model goodness of fit in OLS models. Non-statistically significant terms (*p* < 0.05) shown in brackets. Models fitted with the entire dataset.

Landscape Sizes	Study Areas	Standardised Coefficients										AICc	Nagelkerke R^2^
		MAT	MAT^2^	pNLC	pNLC^2^	LCV	LCV^2^	E	E^2^	Pool	Pool^2^		
5 × 5 km	NYS (n = 4822)	2.12	−2.11	1.19	−1.19	0.20	0.13	0.22	−0.35	0.91	−0.75	12,133	0.28
10 × 10 km	NYS (n = 1075)	−2.53	−2.53	1.41	−1.37	0.00	0.60	0.15	−0.18	(−0.38)	(0.20)	2598	0.35
	ON (n = 985)	1.56	−1.20	2.18	−1.91	0.35	(−0.15)	0.65	−0.36	(0.66)	−0.62	1947	0.58
30 × 30 km	NYS (n = 165)	0.46	−0.47	1.71	−1.58	1.08	−1.13	0.48	−0.52	(0.67)	(−0.84)	373	0.50
	ON (n = 138)	2.62	−2.71	1.05	−0.95	0.29	(−0.34)	0.82	−0.59	(−4.07)	(4.11)	212	0.71

## Data Availability

The Ontario Breeding Bird Atlas (ABBO) data used to obtain presences and absences of the bird species in Southern Ontario are publicly available at: http://www.birdsontario.org/atlas/downloaddata.jsp?lang=en (accessed on 29 April 2017). The New York Breeding Bird Atlas can be accessed through the link: https://www.dec.ny.gov/animals/51030.html. The pNLC covering the study area according to the global 1 km consensus landcover dataset can be found in [37]. Temperature data can be retrieved from [47].

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
