# Peer review of "Avian Diversity Responds Unimodally to Natural Landcover: Implications for Conservation Management"

_animals, 2023, doi:10.3390/ani13162647_

Round 1
Reviewer 1 Report
The authors modelled bird species richness at different spatial scales using quadratic functions. Their findings are interesting and useful for wildlife conservation. Also, authors discussed the pros and cons of their modelling approach, thus offering important insights. Research, analyses and discussion have been finely performed and presented, also being relevant for both the journal and the special issue.
Comments
Lines 2-3 – The title here is different to the title in the mdpi site. I think the latter is more appropriate.
Line 10 – Correct to “as the proportion of natural land cover increases”.
Line 13- breeding birds’ atlas…
Line 13 and throughout – Superscript “2” in km2.
Lines 30-31 and throughout – Please follow the journal style in citing references in the text, e.g., [1-3] here.
Lines 38-41 – This sentence seems incomplete. Please revise.
Line 74 and throughout – It should be Desrochers et al. []. With the relevant reference number in the square brackets. Please advise and follow journal style guidelines.
Lines 83-85 – This sentence is hard to follow. Please revise.
Lines 97-99 – This sentence is hard to follow. Please revise.
Line 105 and throughout – Please follow journal style for subsections.
Lines 153, 162-163, 171-172 – The authors should clearly indicate the years for which data for independent variables were retrieved. They should correspond to the years that bird species richness data was used.
Line 195 – from both methods…
Lines 232-234 – I tried but I could really understand methodology here. Please explain better.
Lines 241-242 – Give reference for R package.
Line 286 – What is “Panel A)” here?
Line 291 – Include site legend in Fig. 4.
Line 296 – LCV
Line 298 - It is p > 0.05 for statistically non-significant terms. Please check and revise.
Table 2 – What is “LC”? Shouldn’t it be “pNLC”?
Lines 340-346 – But diversity and evenness would be more critical for inferring extinction risk. Species richness alone is not sufficient for making such conclusions. The authors recognized this (lines 430-439) but some more in-depth discussion about the usefulness of other indices for detecting potential threats with relevant references should be included here.
Lines 390, 391, 392, 403, 407, 412, 429 – Reference to tables and figures should be avoided in the discussion.
Lines 393-401 – This is results. Revise or move to results.
Some omissions have been detected and highlighted. A more careful reading would improve the quality of the English.
Author Response
I have accepted the vast majority of the reviewer's comments and suggestions.
Thank you for improving significantly the quality of the manuscript.
Regards,
Rafael De Camargo

Reviewer 2 Report
The manuscript is devoted to the problem of assessing the peak diversity of birds depending on the proportion of natural vegetation in the landscape at different scales, using the example of two large geographical units that differ in the degree of anthropogenic transformation and the composition of the avifauna. The authors, through the use of adequate statistical methods and careful preparation of data for analysis, argue for the assumption that most of the differences in landscape richness should be associated with variables other than natural and anthropogenic vegetation cover. The conclusions obtained by the authors give a new vision of the problem of assessing the loss of species diversity in the transformed habitat and allow developing a new approach to biodiversity conservation of biomes.
The disadvantage of the study is that the analysis of the dynamics of the diversity of nesting fauna on a scale of 5x5 km was not carried out for the Southern Ontario region. However, objectively, the authors cannot overcome this shortcoming due to the limitations of the used databases of atlases of breeding birds in this area.
I have no other fundamental remarks. In general, the manuscript can be recommended for publication in the journal Animals. This study may be of interest as a theoretical basis for modeling the cause-and-effect relationships of landscape and biological diversity, as well as for refining predictions of species extinction.
There are minor technical comments on the manuscript, which the authors can easily correct.
1. It is necessary to carefully correct the subtitles of the manuscript.
Page 6, line 182 STATISTICAL ANALYSYS – typo needs to be corrected (Statistical analysis) and change font to lower case Statistical analysis.
Perhaps the italicized subtitles should be numbered rather than not highlighted in special font subtitles such as “Main predictors: the amount of natural land cover, temperature, and landcover variety” (Page 5, line 149) and “Model covariates: sampling effort and the size of regional species pool” (Page 6, line 206) should be in italics.
2. There are typos in the page numbering this manuscript.
3. It is necessary to correct typos in some citations.
Page 2, line 82 (De Camargo and Currie 2015) change on (De Camargo & Currie, 2015)
Page 2, line 123-124 (NYBBA, (McGowan & Corwin, 2008) (McGowan & Corwin, 2008) change on (NYBBA, (McGowan & Corwin, 2008)).
Page 7, line 229 (BirdLife International 2020): Specify the source not in a footnote, but in the list of references.
Page ??? line 411 (Currie 1991) change on (Currie, 1991)
4. It is required to indicate the full number for the R2 values in Figures 3 and 4, Figures in Appendix, as in the text of the manuscript (only the fractional part after the decimal point is indicated).
Author Response

(The authors gave the same response as above.)
